# A Review of the Complex Role of Family Caregivers as Health Team Members and Second-Order Patients

**DOI:** 10.3390/healthcare7020063

**Published:** 2019-04-24

**Authors:** Deborah Witt Sherman

**Affiliations:** Graduate Nursing, Florida International University, Miami, FL 33199, USA; desherma@fiu.edu

**Keywords:** family caregivers, family assessment, interventions

## Abstract

In Palliative Care, the unit of care is the patient and their family. Although members of the health care team often address the family caregiver’s opinions and concerns, the focus of care remains on the needs of the patient. The readiness and willingness of the family caregiver is often overlooked as they are expected to assume a complex caregiving role. When family caregivers are not intellectually or emotionally prepared or physically capable, the caregiver is at high risk for serious health issues and cognitive, emotional, and physical decline particularly as caregiving extends over time. Family caregivers are often a neglected and at-risk population. Illustrated through the use of a case study, this article addresses the complex role of family caregivers, as both health team members and second-order patients. It emphasizes the importance of family assessment and interventions to balance the burdens and benefits of family caregiving and protect caregivers’ health and well-being.

## 1. Introduction

The concept of family has changed over time and has cultural, legal, and sociological connotations. In years past, an individual was born into a traditional nuclear family, with parents, and possibly siblings, and close contact with their extended family of grandparents, aunts, uncles and cousins. Often the nuclear and extended family either lived together or in close proximity, perhaps for a lifetime [1,2]. As women joined the workforce, the care of children or older family member brought challenges associated with caregiving. In addition, the mobility of families, divorce, and travel are social forces that limit the availability of family members to help one another. Furthermore, advances in health care have resulted in increased life expectancy of older family members who may become dependent due to disabling chronic illness. Care of sick children or older adults often becomes a family stressor [3]. The question is “Who stays home to be the family caregiver?” The family caregiver is defined as a spouse, partner, or other family member, who is unpaid, but is responsible for the physical, emotional and often financial support of another person who is unable to care for themselves due to illness, injury, or disability [4].

In the United States, 43.5 million people serve in the role of family caregiving with a third reporting that they have cared for more than one family member [5]. Over 60% of family caregivers are women and the age of the caregiver increases with the age of the care recipient [5]. Not only is there an increasing number of caregivers between the ages of 50 to 75, but it is anticipated that by 2030, there will be over 70 million people age 65 or older who will need caregiving as their independence decreases and symptoms of chronic illnesses increase [5]. Family caregivers often become the “sandwich generation,” as they must care for their children and care for their ill spouse or parents, while still needing to work outside the home for financial support [5]. 

In Palliative Care, health care providers assess patients and their family members experiencing serious, progressive, chronic or life-threating illness and intervene in ways to promote their health and well-being [1]. While the primary focus of the health care team is on the patient’s needs and health, the needs and health of family caregivers may be viewed as secondary, thus the term of second-order patient [1]. In this article, a case study approach is used to examine the phenomenon of family caregiving. “A case study encompasses a problem contextualized around application of an in-depth analysis, interpretation, and discussion, often resulting in recommendations for action or for improving existing conditions” [6]. The review of the literature for a case study paper is focused on providing information which provides background information and enables interpretation of the subject of analysis [6]. Through an introduction to family caregiving, as illustrated by a case study and a review of existing literature, this article addresses the complex role of family caregivers as health team members and second-order patients. It highlights the stressors and benefits of caregiving, the importance of ongoing family caregiver assessment and the need for individualized interventions to promote family caregivers’ health and well-being. The case selected for illustration of a family caregiver experience is based on the author’s unstructured conversation with a family caregiver at a long-term care facility. Verbal permission was given to the author to anonymously document the caregiver’s experience. Names provided in the case are not actual names. The case conveys the author’s interpretation of the experience of Mrs. M and her daughter as the caregiver.

## 2. Case Study

Mrs. M is an 84-year-old Caucasian female with a history of chronic back pain, as well as anxiety and depression. Her husband died when she was 70 years old and, at that time, she was in reasonably good health with no life-threatening illnesses. Mrs. M continued to live in a fifty and older community, not far from her son and two of her three daughters. Her oldest daughter, Sharon, who was a nurse practitioner, oversaw the health care of her mother despite living in another state which was four hours away. Sharon’s sisters, though not working outside the home, maintained their social lives as a priority over the care and assistance they were willing to offer their mother. According to Sharon, their feelings about their mother was rooted in their upbringing with disapproval of her strict parenting style. Mrs. M’s son was available by phone on a weekly basis and perhaps for a monthly luncheon, yet never offered his mother any tangible or practical support.

Over time, Mrs. M’s physical, emotional, and functional abilities declined due to pain, and high psycho-somatic issues. Sharon, who moved even further away for work reasons, was very concerned about her mother’s declining condition. Sharon suggested that her mother move to her state so that she could orchestrate the formation of a health care team to evaluate and treat her mother. With the involvement of her daughters, Mrs. M moved into an independent living facility. Sharon’s younger sister and her husband reassured Mrs. M that they were planning on spending much more time over the year in their second home which was only an hour away. Mrs. M’s children encouraged the move as the weather was much warmer and hopefully, she would have a happy life among people her own age.

With the support of Sharon, who visited weekly, Mrs. M was challenged with beginning a new life in a new state. Within two weeks of her move, Mrs. M was hospitalized for influenza and this began her significant decline in health. At this point, Mrs. M. became highly symptomatic with uncontrolled anxiety and nausea, though her pain was being managed by a pain specialist. The nursing team and physician on site at the independent living community could not address the intensity of Mrs. M’s physical and psychological symptoms. Mrs. M lost nearly 30 lbs. within three months. Her workup included an abdominal CT scan, neurologic evaluation, physical and occupational therapy, and weekly visits to her apartment by a psychologist. Mrs. M’s obsessive-compulsive disorder, depression, and anxiety reached new levels of intensity. Sharon, with good intentions of caring for her mother, was overwhelmed by the number of phone calls from the health care team, visits to the emergency room with her mother and multiple doctor visits. Given that Mrs. M lived alone in an apartment in the independent living facility, Sharon needed to hire private aids for 8 h a day as her mother was unable to care for herself.

Within six months, it was clear that Mrs. M needed to be moved to a health care facility with a full range of services including assisted living, and a nursing home. Sharon asked her siblings to bring Mrs. M back to her home state, but they refused as no one wanted to arrange the transfer, deal with taking her to doctors, or provide the intense physical, psychological and financial support she required. As such, the only and best option would be to bring Mrs. M to the facility where Sharon worked part-time as a nurse practitioner, as she knew the medical and nursing staff and could advocate for her mother’s needs. 

Reluctantly, Mrs. M transitioned from an independent living facility to the assisted living facility. She was not at all happy. Mrs. M did not want to go to the dining room for meals or interact with any of the other residents, refusing to engage in any of the pleasurable activities the facility offered. Within one month of admission, Mrs. M lost an additional ten pounds, reported unrelenting anxiety and nausea, choked on her food and was failing to thrive. Medication management for psychological and pain symptoms offered little improvement. The priest was asked to visit to offer her spiritual support, with the hope that her anxiety would decrease. At this point, Mrs. M could hardly walk, wanted to be fed, and began to have skin breakdown.

Given the stress experienced as a caregiver and the related compromise of her own immune system, Sharon developed ongoing sinus infections requiring serious surgery. According to Sharon, her younger sister and brother continued to take an out of sight, out of mind approach to their mother, fulfilling their duties primarily through phone calls to Mrs. M but never a call to Sharon to offer concern or support. Sharon’s sister, Denise, who was closest to Sharon in age, was concerned about Sharon’s health and visited every three months for two or three days, encouraging her to take a step back from the level of interaction with their mother. However, having a stronger relationship with her mother than her siblings, Sharon was unable to step away.

Within two months of coming to the assisted living facility, Mrs. M was enrolled in home hospice. Mrs. M had 4 h of care provided by a hospice aid, and 8 h of private care arranged by Sharon. When in the apartment alone, Mrs. M incessantly pressed the call button to the front desk of the assisted living facility, screaming “Help me, somebody help me.” Yet, this was Mrs. M’s mantra, even when a nursing aid was at her side offering assistance. The diagnosis was early stage dementia with behavioral disturbance, while the psychologist believed that her behavior was more reflective of a previously undiagnosed personality disorder, as well as anxiety and depression. 

One night, Mrs. M had an acute change in cognitive status and inability to be aroused. Sharon was called and Mrs. M was taken to the hospice respite unit. Within two days, she woke up and, with encouragement of the private aid, she began to eat. Mrs. M was discharged from the hospice respite unit to a bed in the nursing home in the medical facility rather than returning to her apartment in assisted living. All medications except her fentanyl patch were discontinued and she could be heard screaming from the minute her daughter came off the elevator. Clearly, the pain and symptom management offered by the hospice team was insufficient. Sharon, upset by the discontinuation of all of Mrs. M’s pain and psychiatric medications, decided to call in the pain expert of the facility. According to Sharon, this was viewed upon negatively by the director of Hospice, who knew that she was a nurse practitioner, and showed little regard to the distress and frustration she was experiencing as a daughter.

Within the week, Mrs. M was abruptly discharged from Hospice care, citing that she had a prognosis of more than six months to live. A health care team, comprised of the facility’s primary care physician, nurses and advanced practice nurses and psychiatrist were assigned to her care. With complaints about Mrs. M’s screaming, she was admitted to a dementia unit of the nursing home. Sharon was outraged as she did not feel her mother was receiving coordinated, effective care, yet her hands were tied. 

As occurred previously, the medical and nursing staff called Sharon frequently regarding her mother’s condition or her behavior. Sharon was treated as a health team member, rather than recognizing that Sharon was clearly becoming a “second-order” patient in need of care and support herself. At this point, Sharon developed hypertension, insomnia, and began to feel consumed by the caregiving experience and in a constant state of anxiety. Sharon had two episodes of pneumonia and was hospitalized during the last episode for two weeks. 

Two of Sharon’s siblings felt that even hearing about their mother challenged their emotional well-being. If they could reach Mrs. M by cell phone, they would. Otherwise, they did not call Sharon with true concern over their mother’s condition or how their sister was coping. In fact, Mrs. M’s son visited his mother once a year though he was frequently in the state on business. During his last visit, when his mother asked when she would see him again, he responded “I’ll see you in heaven,” openly expressing that he prayed for her “to die and be taken to heaven.” For all essential purposes, Sharon, as a caregiver, felt abandoned by her younger sister and brother. 

Sharon continued to suffer from her own feelings of remorse in bringing her mother to her state as no matter what support she offered, Mrs. M just wanted to return to her extended family. Given her own self-imposed feelings of responsibility, Sharon visited her mother each day she was at the facility in her role of health practitioner, and she often took her mother “off campus” to “engage her in the land of the living,” despite her mother’s frequent request to “let me die.” 

Unfortunately, in many family situations, the illness and death of a family member can create extremely strained family relationships, particularly when there is a perceived lack of support by the family caregiver. Sharon, who is on the front line of her mother’s care, finds comfort in knowing that she is doing everything possible to offer love, care and support, even though it significantly heightened her own health risks and compromised her quality of life. 

As happens with so many family caregivers, Sharon has shouldered the daily burden and has become physically and emotionally depleted. Interventions are needed to offer her support and comfort, but she is viewed in the role of health care provider by her family and the health care team, not as a distressed daughter. Perhaps, in wearing her white coat, this is Sharon’s shield from her personal pain and distress. This case study illustrates that family caregivers are expected to be a member of the health care team, when, in reality, they become second-order patients, with little attention given to their needs and concerns from people both within and external to the family. In too many cases, the family caregiver is a member of an at-risk, invisible, vulnerable population.

## 3. Caregivers Are Expected to Be Members of the Health Care Team

As illustrated by the case of Mrs. M, the family caregiver role begins immediately at the point of the patient’s diagnosis and continues across the illness experience as the disease progresses [7,8]. Family caregivers may live with the ill individual, serving as the primary caregiver, or may live separately from the person receiving care. The average number of hours of caregiving per week is 20.4 h, with women spending greater time caregiving then men (21.9 vs. 17.4 h/week) [5]. Furthermore, with the increase in Alzheimer’s dementia, the role of the caregiver often extends to greater than five years. The role of the family caregiver may involve bathing, dressing, feeding, or toileting, house cleaning, shopping, cooking, taking the family member to medical appointments, serving as the medical interpreter, offering emotional and spiritual support, as well as assistance in managing the finances [9,10]. 

Today, in addition to all of these roles and responsibilities, with the increasing diagnosis of chronic, life-threatening illness and extended life of patients with advanced disease, the family caregiver must also identify and treat disease-related symptoms [5], administering many medications, observing for side effects, and monitoring for drug interactions. With minimal preparation or training, family caregivers are often required to care for family members with feeding tubes, urinary catheters, on ventilators with tracheostomies, receiving dialysis, or they may be responsible for administering intravenous medications, or changing wound dressings [10]. Often family members are not asked if they are willing to be family caregivers for their ill or disabled family member, but they are expected by the health care team to assume the role upon hospital discharge or care for the patient in the community setting. 

Health professionals view family caregivers as a member of the health care team when they: (1) ask caregivers for all or supplemental information about the patient; (2) ask for information about the patient’s symptoms or cognitive and physical functioning; (3) ask if the medications prescribed are effective; (4) discuss with them the patient’s problems and treatment options; (5) ask them to interpret their conversation and recommendations with the patient [11]. In fact, caregivers become not only a conduit for information between patient and provider, but with the extended family [12]. 

Without family caregivers, patients’ survival rates are lower and societal costs for end-of-life care are greater [13], and patients are placed in more costly hospital or nursing home settings [14]. In addition, they are at greater risk for poor care or neglect [15,16]. The concern is that as patient’s performance status declines over time [17], negative caregiver outcomes may limit optimal care [15]. It is documented that caregiver well-being is closely linked with patient well-being [18,19,20]. Sautter et al. [21] reports that the burden of caregiving occurs early in the course of advanced illness and, therefore, early screening and intervention for caregivers should not wait to the end of life. Caregivers often have a less hopeful outlook than patients, which suggests an unmet need for education, communication, and support services tailored to their caregiver needs. 

Furthermore, when ill family members have very aggressive cancers, family caregivers are distressed by the very rapid illness trajectory, the few treatment options, and the management of the patient’s intense symptoms [22]. The family caregivers have limited time to prepare for or adjust to these fatal diagnoses, and experience the full range of emotions, including uncertainty, depression, anxiety, fatigue, and grief, and worry about their own genetic risk for the disease [11,23]. In contrast to a rapid illness trajectory, for patients diagnosed with dementia, the skills needed for effective caregiving must be learned over time as the demands and stressors associated with caregiving increase from the early to the advanced stage of the disease [1]. As the caregiving experience is prolonged, the potential for compromise of the family caregiver’s health and quality of life is increased as illustrated in the case study of Mrs. M. 

## 4. Family Caregivers Become Second-Order Patients Given Intense Distress

Despite the National Consensus Guidelines for Quality Palliative Care [24], which emphasize that both patient and family should be viewed as the unit of care, health professionals do not intervene in ways that take caregiver well-being into account. Furthermore, resources for caregivers are limited, fragmented, and discontinuous, despite the evidence that family caregivers experience intense physical, emotional, and social distress [25]. 

Physical stress and strain may occur as family caregivers assist patients with walking, lift heavy wheelchairs into cars for appointments, assist the ill family member into or out of bed or lift them if they fall. Physical strain also is exacerbated by lack of sleep which occurs due to stress or when caregivers are awakened by the ill person. In turn, the caregiver experiences reduced energy levels and a decrease in their own physical function [7]. Overwork of carrying out all other family responsibilities exhausts the physical reserves of family caregivers [1]. 

The demands of caregiving create significant emotional stressors, associated with fear, confusion, powerlessness, grief, and a sense of vulnerability, despite attempts to maintain normalcy [7]. Caregivers suffer from symptoms of anger, depression, and anxiety, resulting in a sense of demoralization and exhaustion [10,23]. Additional stress also occurs when family caregivers are asked to make treatment decisions for their ill family member, especially if they have not had previous conversations about goals of care and preferred treatments [1]. Many caregivers of advanced cancer patients demonstrate impaired cognitive functioning [26], and are treated for psychiatric problems [27]. Both the physical and emotional stress of caregiving often leads to new physical illness of the caregiver, exacerbation of co-morbidities, and a greater risk of caregiver mortality. Although there may be increased caregiver distress, caregivers are often reluctant themselves to shift attention away from the patient. [7] Some caregivers hide their feelings of loss and grief from patients [28,29,30], leading to increased isolation, depression and risk for complicated grief during bereavement [28,30].

Socially, caregivers experience altered household and family roles and communication patterns [31]. As caregivers abandon leisure, religious, and social activities, there is heightened marital and family stress, with long-term consequences for the family [32]. Burdens related to time and logistics, and lost wages or leaving the workforce entirely, have severe personal, economic, social, and institutional implications for the family caregiver and the entire family [33,34].

According to Given et al., [31] caregiver burden is a multidimensional bio-psycho-social reaction that results from an imbalance of care demands that are relative to caregivers’ personal time, social roles, physical and emotional states, and financial resources, and other role responsibilities. Health risks and serious illness of caregivers may increase their utilization of health care resources, contribute to escalating health care costs, and place caregivers at greater risk for life-threatening illness [5]. As such, family caregivers are a vulnerable and at-risk population who are largely neglected by the health care system [5]. Targeted interventions for family caregivers are critically needed as caregivers are second-order patients [35]. Certainly, as in the case study, Sharon, the family caregiver, experienced many of the stressors described above and needed the health care team to offer care and support.

## 5. Understanding the Changes and Transitions Associated with Caregiving

Examining changes and transitions that occur in the broad stages of a disease-centric model are inadequate to identify the inherent changes, compressed time-points, and key transitions experienced by family caregivers with unique needs during illness and bereavement [36]. Life-threatening illness is a disruptive event which creates change with respect to daily adjustments and associated transitional events. Caregiver transitions encompass the patient’s phases of illness, as well as changes in health care providers, and location of care [37]. In the care of patients with cancer, the key transitional experiences have been reported as the start of treatment, end of treatment, non-routine hospitalization, leaving the hospital for home, transitioning from curative to palliative care, entering hospice, patient death, and during bereavement [37]. Longitudinal studies identify the specific, incremental needs of family caregivers across the illness experience [36]. Results indicate that family caregivers experience significant physical and emotional symptoms, and lower quality of life as the patient’s illness progresses and caregivers must change and adapt continuously to new patient needs [38]. “Characterizing the progression of multiple dimensions of caregivers’ experience from serious illness through death of the patient is crucial to an evidence-based understanding of improving end of life care and to capturing transitions and end of life trajectories” [39]. The case study highlights the many transitions experienced by Mrs. M, in terms of the phase of the illness, along with changes in health care settings and the associated impact on both her and her daughter as the caregiver.

## 6. The Silver Lining in Family Caregiving: Perceptions of Benefits

Caregiving for seriously ill and dying family members can also be perceived as a positive experience. Kang et al. [40] stated that some of the positive consequences include personal growth, satisfaction and sense of accomplishment, strengthening of relationships, and a change of worldview. Family caregivers have listed feelings of reward from a set of different circumstances, such as being helpful to the patient, bringing happiness to the patient, making life easier for the patient, and just being there [41].

The positive aspects associated with the caregiving experience may act as a buffer against overwhelming burden and traumatic grief [23,25,42,43]. Caregivers who have a positive approach to life are better able to cope with caregiving demands [44] and are motivated to maintain their caregiving role [45]. Funk et al. [7] report a sense of existential meaning associated with the caregiver role, including a sense of pride, esteem, mastery, and accomplishment. Spousal caregivers shared feelings of enhanced relationships, reward, personal growth and satisfaction and even benefit in bereavement [46,47]. Even adult children of cancer patients report benefits of transformed relationships with sick parents, valuing the relationships with other family members, altered life priorities and personal development [48,49].

Religious beliefs, age, gender, and socioeconomic factors are associated with the perceptions of benefits of caregiving. The caregiver experience is more positively perceived by caregivers who have a strong religious faith [40]. In addition, women are more inclined to express appreciation for the experience of caregiving with a greater sense of connection to others [40]. Older caregivers also perceive benefits as they understand more fully the meaning of life and view traumatic events as less stressful than younger caregivers [40]. In many situations, caregivers may view themselves more positively with an increase in a sense of self-efficacy as they handle successfully stressful situations [50].

Resilience is a personal characteristic which increases caregivers’ ability to care and play a protective role [51]. Caregivers’ resilience is increased when they perceive a sense of support from other family members, friends, and the community. Strong social networks and socio-economic factors such as education, housing status, and employment status are also related to caregivers’ resilience [52]. When adversity is experienced related to illness or the experience of caregiving, caregivers may actually experience what is termed as post-traumatic growth [53], which leads to a revaluation of life and its values and an affirmation of important relationships. Post-traumatic growth due to adversity also supports the caregiver’s transition into the bereavement period. In the case study presented, Sharon expressed hope that upon her mother’s death, she would be able to reflect on the positive aspects of caregiving, with less emphasis on caregiver burden.

## 7. Caring for the Caregiver Begins with a Family Assessment

A basic tenet of the first domain of the National Consensus Project for Quality Palliative care (NCP) [23] is that the patient and family are the central focus of the interprofessional health care team. Family assessment is, therefore, critically important to understanding family member’s needs, concerns, and perceptions, as well as those of the patient. A family assessment begins with obtaining demographic information. Questions include the caregiver’s relationship to the patient, do they live together or separately, who is in the household, what is their financial status, and what is the employment status of the patient and caregiver [5]. 

It is also extremely important to understand the cohesiveness of the family and the family dynamics. Additional questions include: Who is seen as the head of the family? Who makes decisions regarding the family? How do family members communicate? Is the family open to others outside of the family? What are family cultural values and expectations regarding caregiving? Is the caregiver able and willing to provide care? Is the care recipient willing to receive care? How has caregiving affected the caregiver’s life and health? Is there distress in the caregiver’s voice, or does their thoughts and behavior indicate anxiety, depression, anger, hopelessness or despair? What plans of the caregiver are on hold as they care for the patient? And what help is needed by the patient and family caregiver [5,11].

It is important to obtain the answers to these questions over time, in the presence of the patient and through planned separate conversations with the family caregiver. As the patient’s health deteriorates, it is equally important to continue to ask the caregiver about their health and well-being, acknowledging the burdens and supporting their caregiving efforts. In the case study, Sharon, as a caregiver, expressed her appreciation of sharing her experience and consideration of her concerns and needs.

## 8. Providing Individualized Caregiver Interventions Rather than a One-Size Fits All Approach to Care

A one-size fits all approach to the care of family caregivers does not take into account differences in the experiences of family caregivers based on age, gender, race, relationships, or caregivers’ coping, as well as needs and preferences for interventions [52,54]. Further, caregivers’ relationships with patients and their adjustment may indicate vulnerability to challenges of caregiving and may inform interventions that reduce stress and increase personal growth [10,49].

Interventions need to be developed to bolster positive caregivers’ coping, and resources, with a focus on the benefits/gains, meaning making, and possible personal growth related to caregiving. In addition to the attention a health provider shows to the patient, it is equally important to greet the caregiver and include them in the conversations while making eye contact and offering supportive gestures. Listening acknowledges the concern, needs, and stress of both the patient and caregiver. Honesty can also be gained when the caregiver is given the opportunity to have a private conversation with the health care provider. This may be the time when the caregiver feels free to share the burdens and benefits of caregiving and their needs and preferences for support can be determined [11].

Based on a comprehensive ongoing family caregiver assessment across the illness trajectory, health care professionals can determine the right type of caregiver intervention to be provided at the right time, in the right dose. A web-based intervention psycho-social intervention, called the Comprehensive Health Enhancement Support System (CHESS), was shown to have a modest effect on reducing caregiver burden (d = 0.387) and caregivers’ negative mood (d = 0.436) at six months [55]. Similarly, an educational intervention offered to patients experiencing prostate cancer and their family caregiver was shown to increase patient’s and partner’s cognitive, problem solving, and behavioral coping skills with significant improvement in emotional distress and sexual function within three months of follow-up [56]. A systematic review performed by Waldron, Janke, Bechtel, Ramirez and Cohen [10] suggests that interventions targeting communication and education have an impact on improving caregivers’ quality of life, promoting a more optimistic attitude regarding caregiving, and reducing feelings of uncertainty and fear. 

It is also important to assess when there is a need to escalate an intervention by referral of the caregiver to other members of the interprofessional team or to have a multi-focal approach. When the family caregiver is becoming overwhelmed, discussion of home health care or the support offered by hospice is extremely important to preserving the well-being of family caregivers. Targeted strategies to support family caregivers also includes having difficult conversations, setting realistic goals, negotiating expectations, and finding help [57]. It may be difficult to realistically discuss the patient’s diagnosis and prognosis, and understand the underlying dynamics of the patient and family, which can either be positive or negative depending on their history. 

Setting realistic expectations and goals are important and necessitates an understanding of what the patient is capable of or not capable of doing due to cognitive or physical deficits. Caregivers must also set realistic expectations of themselves and negotiate with other family members competing priorities. Negotiating expectations involves helping family caregivers to let go of some responsibilities and releasing themselves from feelings of guilt that they are not doing enough. A sense of despair and hopelessness of the family caregiver, as well as lack of support and acknowledgement of the caregiving role place the patient at increased risk for abuse [7,23,30]. Finding help may begin within the family system and extended family, as well as the patient-caregivers’ community. Help is also available through support groups, and social programs and organizations such as the Family Caregiver Alliance [5] or the National Alliance for Caregiving [4]. Yet, difficulties exist when caregivers are not open to outside assistance and support for themselves as they feel that do not have the time to participate. In this situation, web-based interventions, which used the Internet for delivery, such as caregiver forums, online support groups, virtual communities, smart phone applications or on -line programs offering caregiver information may be successful options [55].

Family caregiver interventions need to: (1) be based on caregiver data obtained through longitudinally designed studies; (2) promote the benefits of caregiving, and meaning making, which enhance caregivers’ personal growth; (3) consider perceived needs and preferences of caregivers depending on coping styles, and interest in the use of web-based technology; (4) be developed depending on the presenting issues of patients and caregivers; (5) intervene in supporting relationships of patients-caregivers-health professionals; (6) identify sub-groups (age, gender, relationships etc.) in tailoring interventions; (7) be flexible, integrated, multi-faceted, and individualized; (8) be “dosed” based on the degree of baseline and progressive caregiver distress; (9) offered at times perceived as key time-points and transitions experienced by caregivers; and (10) be based on clearly identified and measured caregiver outcomes [58,59,60,61,62]. Mrs. M’s case study illustrates the importance of assessing the well-being of family caregivers. It encourages health professionals to address the needs of family caregivers and provide individualized interventions, if family caregivers are to maintain their own health and continue to provide love and support to their ill family members.

## 9. Conclusions

The role of the family caregiver remains two-fold. One is as a member of the health care team who cares for the patient, and secondly, as a member of the unit of care whose needs, concerns, and health must be addressed. Health team members need to be ever aware that a caregiver may give care with all good intentions until they begin to succumb physically, emotionally, socially or financially to the intense complex stressors of the caregiving role. At that point, the team focus must pivot to acknowledge the caregiver as second-order patients and to help family caregivers who are at-risk for morbidity and mortality. When caregivers are unprepared to provide instrumental, emotional, and financial support, caregiver risk escalates [58]. When caregivers feel pride in their care, have sufficient support from family and health professionals, and believe that their role increases their sense of meaning and purpose in life, they approach their role with hope and optimism, and caregiver risk decreases [46]. An interprofessional approach to the care of family caregivers is extremely important to limiting the burdens of caregiving and promoting the perceived benefits of the experience. Continual assessment of caregiver burden and benefits are two important concepts to understand within the context of patient-family centered care.

Health care professionals need to prevent and reduce caregivers’ illness and exacerbation of co-morbid conditions; reduce caregiver burden; enhance the benefits of the caregiving experience; evaluate and treat physical and emotional symptoms of caregivers; support the personal growth and the quality of life for family caregivers across the illness trajectory and bereavement; and advocate for health care resources to support family caregivers [57]. Although the health care system relies on family caregivers to offer physical and emotional care to the ill family members, family caregivers must be recognized as “care recipients,” with a right to their own support, and fulfillment of their needs and to have their experience evaluated “not as a proxy response for patients but as an outcome itself” [39].

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
