# Peer review of "A Review of the Complex Role of Family Caregivers as Health Team Members and Second-Order Patients"

_healthcare, 2019, doi:10.3390/healthcare7020063_

Round 1

Reviewer 1 Report

I appreciate the opportunity to review this paper. I believe the topic to be timely and important, especially the focus on involving caregivers in care. However, I do have several major concerns. I will provide them below and hope the information will contribute in a positive way.

Methods: First, from a methodology perspective, it is not clear how the case study or narrative was collected and if permission is granted to share this story. Please explain the methods and how this story was acquired. There is significant detail that seems to be very personal for the caregiver but also that person’s family, and has a tone which suggests frustration with certain family members (language such as abandoned). Though this is often the case, it is not clear if these were the individual’s words or the interpretation by the author. Also, it is not clear that those family members would know that the story is being released and that seems unethical to share such a story without their permission as well as clear indication of how a case study/narrative was collected. I would suggest a clear methodology section. This also relates to the title suggesting it is a review – I suggest following review protocol for collecting and summarizing articles in a systematic way.

Organization: Other suggestions include re-organizing the paper to flow a bit from section to section to make connections and build for an overall purpose. Sections focus on important topics in caregiving but it is not fully developed or connecting or building across the paper. Also the literature could be developed further as there are more research or studies that are not included in the background information as well as updating citations and some grammar corrections. Addressing the methodology for a review will help with gaining more literature, or possibly refining the focus to be specifically on one aspect which is key such as the involvement of caregivers (as these sections were indeed strongest).

Again, it seems the author seems to be particularly interested in the connection of caregivers in the clinical context and might want to focus on that component more specifically including the story. 

Author Response

Thank you for the opportunity to revise the manuscript titled “Review of the Complex Role of Family Caregivers as Health Team Members and Second Order Patients.’ I greatly appreciated your important feedback which has strengthened the article for publication in Healthcare.

Response to Comments of Reviewer #1

1.       REVIEWER # 1: Content was included which discusses the use of a case study approach in examining the experience of family caregiving. As in clinical care, case studies are used to exemplify a phenomenon or situation rather than following the research protocol of a case study. To enhance the literature of family caregiving for health care providers, a case study approach was therefore used.

2.       REVIEWER #1: Clarification was provided as to the permission of the family caregiver for the author to share her experience with the identity of the caregiver as anonymous.

3.       REVIEWER #1: In the text, it was indicated that the author was informed about the case through several unstructured conversations with the caregiver who she met informally in the nursing home of the health care facility.

4.       REVIEWER # 1:  It was stated in the text that the case study was an interpretation of the author rather than the explicit words of the caregiver.

5.       REVIEWER # 1:  The flow of the article was reviewed and references were made to the case study to illustrate the caregiver experience within the context of the review of literature of family caregiving such as: expectations of being a part of the health care team, caregivers becoming second order patients due to intense distress, transitions associated with caregiving, benefits of caregiving, caregiver assessment and the need for individualize caregiver interventions. The topical outline followed was consistent with the focus of the paper. Although this article provided a review of the literature as stated in the title, the work was not conducted as an integrative review or a systematic review with clearly specified methodology. The literature cited contained many references published within the last five years, as well as seminal articles related to family caregiving.

Reviewer 2 Report

Please add the information identified in the text

Author Response

Thank you for the opportunity to revise the manuscript titled “Review of the Complex Role of Family Caregivers as Health Team Members and Second Order Patients.’ I greatly appreciated your important feedback which has strengthened the article for publication in Healthcare.

Response to Comments of Reviewer #2

1.       REVIEWER #2 responded to the sentence "In Palliative Care the unit of care is the patient and family." The reviewer’s recommended statement is "In Palliative Care, units of care are patients and families." This is not how we write it in our Palliative Care textbook or in the National Consensus Project for Quality Palliative Care.  So I prefer to keep the original sentence.

2.       REVIEWER #2: The suggestion was to include methodology related to the case study. As a response, content was included in the article which discusses the use of a case study approach in examining the experience of family caregiving. As in clinical care, case studies are used to exemplify a phenomenon or situation rather than following the research protocol of a case study. To enhance the literature of family caregiving for health care providers, a case study approach was therefore used. Clarification was provided as to the permission of the family caregiver for the author to share her experience with the identity of the caregiver as anonymous

3.       REVIEWER #2:  It was suggested in section 2- caregivers as health team members- to include a statement regarding the stress of decision making for caregivers. The following sentence was added in response to the reviewer’s comments.  In contrast to a rapid illness trajectory, for patients diagnosed with dementia, the skills needed for effective caregiving must be learned over time as the demands and stressors associated with caregiving increase from the early to the advanced stage of the disease {1]. As the caregiving experience is prolonged, the potential for compromise of the family caregiver’s health and quality of life is increased as illustrated in the case study of Mrs. M.” 

·         REVIEWER #2: It was suggested to refer to the stress of decision making for caregivers. The following sentence was added under the section that caregivers become second order patients. Additional stress also occurs when family caregivers are asked to make treatment decisions for their ill family member, especially if they have not had previous conversations about goals of care and preferred treatments [1].

Reviewer 3 Report

Manuscript Title under Review:  A Review of the Complex Role of Family Caregivers as Health Team Members and Second Order Patients

Journal Submission: Healthcare Editorial

Authors used a case study to highlight how family caregivers are second order patients, who experience physical and/or mental health challenges as a result of ongoing and long-term caregiving demands. The case study refers to Mrs. M who is primarily cared for by her nurse practitioner daughter who transitions Mrs. M from living alone to living in an independent living facility in a new state closer to her daughter, immediately following, Mrs. M was hospitalized due to “influenza and this began her significant decline in health”. Mrs. M moved into several different facilities and her daughter experienced challenges with her health and the complex system of health care providers. Authors presented a review of the literature and recommendations for addressing the needs of family caregivers who are often presumed to be part of the healthcare team and often overlooked as part of the client system.

Major Comments:

·       Case Study: In the case study, I noted comments in the PDF file about the caregiver’s daughter. It was unclear at times if Mrs. M stayed in the same facility where her daughter was employed. If she did stay at the same facility where her daughter was employed, I think her daughter’s role would have been very confusing and conflicting. I state this because not only is she Mrs. M’s daughter, but she is her caregiver, and she is employed there as a service provider. So, how does she serve as a caregiver and service provider to her mother? I think this needs to be addressed or at least noted in the case study for two reasons. The first reason is it IF the caregiver stayed in the same facility where her daughter (caregiver) was employed. Her multiple roles (daughter, caregiver, and service provider) created a very complex caregiving situation. The second reason is the concept of second order patient as defined by authors, “family caregivers are expected to be a member of the health care team, when in reality, they become second order patients, with little attention given to their needs and concerns from people both within and external to the family”.  So, this daughter, if worked at the same facility was in a dual role of service provider and caregiver to her mother. I think if authors clarify the daughter’s role as it relates to these various institutions she was in, it would be clearer.

 Case Study: The case study is a very good case and demonstrates the complex care provided by caregivers, contact with different service providers, and challenges face. Because it is so complex, it was difficult to follow at times. Mrs. M was living alone, then she moved to her daughter’s state to an independent living facility but then soon after she became sick and was hospitalized, and then from there, she was discharged back to the daughter’s apartment (?). I assumed she was because authors noted: “Her daughter could not leave Mrs. M alone in her apartment and needed to hire private aids for 8 hours a day.”  My point is there were several transition points throughout the case study, it was difficult to follow all of Mrs. M’s transitions in care. Possibly a chart or conceptual map might help with tracking all the changes and make the case, “reader friendly”.

Case Study: Authors noted Mrs. M entered hospice care, So, is she in hospice care at home, in her daughter's place or a hospice care facility? and if it is a facility, is it the same facility where her daughter works? It is unclear. Authors need to clarify this area.

Second Order Patients: Authors make a very valid point about second order patients and make a strong case for this with the case study, but it is not until page 3 that the word “second order patient” is introduced. I recommend this concept is introduced at the beginning of the manuscript and one paragraph is included to describe what is second order patients. I know the authors describe this later in the document but would like to some discussion of this upfront.

Minor Comments:

·       Case Study (page 2, Para 3): This case is so complex with so many members, it was hard to follow. Is this practitioner daughter the caregiver daughter? If so, give her a fictitious name, such as Mary or something

·       Page 2, Para 3, Line 14:  change “were” to where

·       Page 3, Para 4, Line 2: from, instead of "for"

·       Page 4, Para 4, Line 1: unclear sentence structure "aggressive and cancers". You may have meant “aggressive cancers”

·       Page 7, Para 2, Line 3: please write CHESS for the first time, then use an acronym. Comprehensive Health Enhancement Support System (CHESS)

Author Response

Thank you for the opportunity to revise the manuscript titled “Review of the Complex Role of Family Caregivers as Health Team Members and Second Order Patients.’ I greatly appreciated your important feedback which has strengthened the article for publication in Healthcare.

Response to Comments of Reviewer #3

1.       REVIEWER #3: The term “second order patient” was explained in the article’s introduction as requested.

2.       REVIEWER #3: The case study was significantly revised, as highlighted in red, to identify the family caregiver by the pseudo-name, Sharon. Clarification was further given to her role as family caregiver, not in the role as a formal member of Mrs. M’s health care team, despite her professional experience as a nurse practitioner. 

3.       REVIEWER #3: Edits were made in the case study to clarify changes in care settings and associated transitions experienced by Mrs. M to make it easier for readers to follow the evolving illness and caregiving experience.  It was stated that hospice care was provided in the assisted living setting where Mrs. M resided in her own apartment and that the health care facility included all levels of care including assisted living, hospice respite care unit, and a nursing home. With greater clarity, the author believed it would not be significantly helpful to include a chart or map to track changes in the caregiving experience.

4.       REVIEWER #3: Minor edits were made as recommended regarding change from “aggressive and cancers” to “aggressive cancers.” However, given line changes, the author could not identify the reviewer’s requests to make grammatical changes such as “were” to “where” or from “instead of” to “for.” Perhaps in the editing of the case study, these requested edits could not be identified. CHESS was spelled out as Comprehensive Health Enhancement Support System (CHESS) as requested.

5.       OVERALL: The format of the references in the text was changed to [1] or [53,54} or [56-62] as requested.

I hope that with further elaboration regarding the author’s response to reviewer’s comments that all requests have been thoughtfully addressed.

Round 2

Reviewer 1 Report

I do believe this paper is improved, and I appreciate the author’s work to address concerns and be responsive to comments. I also think the focus and clarity of second order patients is helpful and clear as the main point and is a contribution. There are still some concerns remaining that I do believe should be addressed (and can be addressed), including:

1.       It was helpful to see added language on permission but lines 54-57 could be further clarified and edited (such as concretely clarifying permission from the caregiver and the form – verbal or written with written preferred). Is it more accurate to say the case as written is based on unstructured conversations with the family caregiver (that occurred where?) and relays the care reason and caregiver experience – this is the language from the reviewer response page.  

2.       There are several statements that refer to the siblings (or others) and their thoughts and perceptions – given these individuals were not interviewed/talked to (nor asked permission) I believe this needs revision or exclusion, including clarifying that this was as reported by the caregiver. Such as lines 64-68 – “Their feelings…” and well as the later mention of their sobriety being challenged. Though relevant, particularly when referring to the siblings there seems to be a level of information beyond what should be reported without their permission (even though anonymous and from the caregiver perspective). Objective language such as: “as reported by the caregiver” would help or, again, remove or modify/temper and still make the point of being an isolated caregiver without the personal detail of those not involved in this reporting. Similar comment for 108-110 lines – According to the caregiver/sharon, siblings remained less or non-involved only with calls or some…similar for lines 128-129 regarding the views of the hospice director. Suggest also removing lines 155-162 for similar reasons but also the next paragraph is helpful in making the point.

3.       I assume names provided are not actual names and making note of this would be helpful or possibly use caregiver and patient if feasible (see Bevans JAMA article).

4.       Writing is thorough (especially after the case report) and improved, but I believe it would still benefit from further/final review for some phrasing or word choice at times – e.g., lines 24 and 25 – “become an issue.” Also noting the caregiver definition – notes spouse and partner then family member – modify to “other family members” would be better? Also in the abstract – using “vulnerable” and “at-risk” together (might these be considered the same?). Also, line 87 – “barraged” is strong.

5.       Not all references are all accessible from the information provided in the reference list unless I am missing it – e.g., #5 and #4 for example. This did not allow to check the sources (for example for paragraph #2 that uses reference #5 throughout and later in the text – was this their research or a summary and in what year for the statistics such as stating number of caregivers nationally (since it is from 2013 state the year in which the data was collected – or consider other recent reports such as NAC/AARP report from 2015).

6.       A minor suggestion in readability, I wonder if section 5 of the paper is needed in terms of flow, or can it be transitioned differently? Other sections built well and related back to the case study.

I hope these comments are viewed as helpful, and, again, the focus on recognizing the emotional needs of caregivers is a clear point and in general the case study is helpful in making the point, but I do think these points should be considered before moving forward.
